# 3D modeling of vector/edge finite element method for multi-ablation technique for large tumor-computational approach

**Gangadhara Boregowda**[ORCID]◎*, **Panchatcharam Mariappan**◎

Department of Mathematics and Statistics, Indian Institute of Technology Tirupati, Andhra Pradesh, India

◎ These authors contributed equally to this work.
* ma19d502@iittp.ac.in

**Data Availability Statement:** All relevant data are within the paper and its Supporting information files.

**Funding:** The author(s) received no specific funding for this work.

## Abstract

Microwave ablation (MWA) is a cancer thermal ablation treatment that uses electromagnetic waves to generate heat within the tissue. The goal of this treatment is to eliminate tumor cells while leaving healthy cells unharmed. During MWA, excess heat generation can kill healthy cells. Hence, mathematical models and numerical techniques are required to analyze the heat distribution in the tissue before the treatment. The aim of this research is to explain the implementation of the 3D vector finite element method in a wave propagation model that simulates the specific absorption rate in the liver. The 3D Nedelec elements from $H(curl; \Omega)$ space are used to discretize the wave propagation model, and this implementation is helpful in solving many real-world problems that involve electromagnetic propagation with perfect conducting and absorbing boundary conditions. One of the difficulties in ablation treatment is creating a large ablation zone for a large tumor (diameter greater than 3 cm) in a short period of time with minimum damage to the surrounding tissue. This article addresses the aforementioned issue by introducing four antennas into the different places of the tumor sequentially and producing heat uniformly over the tumor. The results demonstrated that 95.5% of the tumor cells were killed with minimal damage to the healthy cells when the heating time was increased to 4 minutes at each position. Subsequently, we studied the temperature distribution and localised tissue contraction in the tissue using the three-dimensional bio-heat equation and temperature-time dependent model, respectively. The local tissue contraction is measured at arbitrary points in the domain and is more noticeable at temperatures higher than 102˚C. The thermal damage in the liver during MWA treatment is investigated using the three-state cell death model. The system of partial differential equations is solved numerically due to the complex geometry of the domain, and the results are compared with experimental data to validate the models and parameters.

## Introduction

Ninety percent of cancer-related deaths are caused by hepatocellular carcinoma, which is the fourth most common type of liver cancer [1]. The best therapies for hepatocellular carcinoma patients are liver transplantation and surgical procedures since they reduce the chance of

**Competing interests:** The authors have declared that no competing interests exist.

developing new tumors [2]. Oncologists advise minimally invasive procedures such as local ablative techniques and liver resection due to patients' habits, the location of the tumor, and a lack of adequate donors. For the treatment of cancer, a variety of ablative methods have been employed, including cryoablation, radiofrequency ablation, and microwave ablation [3]. Cryoablation is a form of thermal ablation that involves liquid nitrogen or argon to freeze unwanted cells during the treatment [4]. One of the most used thermal ablation methods for treating cancer is radiofrequency ablation, which employs radio waves to create heat inside the tumor [5]. Microwave ablation (MWA), a local ablative technique, uses microwave energy to kill unwanted cells. During MWA, the antenna is inserted inside the liver to transfer the microscopic energy at a frequency of 2.45 GHz. A single-slot coaxial antenna with a slot size of 1 mm was used to produce adequate heat in the liver [6]. Rapid and uniform heating, higher intramural temperatures, reduced susceptibility to heat sinks, a shorter ablative duration, and a bigger ablative volume are benefits of MWA over radiofrequency ablation [7, 8].

Since microwave ablation can generate extreme heat around tumor cells and damage neighboring healthy cells, it presents a significant difficulty. The effect of heat deposition and thermal damage around the tumor can be predicted before treatment with the aid of mathematical modeling and computation tools. The wave propagation model obtained from Maxwell's equation is used to simulate the electric field intensity in the liver for an input power of 50 W and a frequency of 2.45 GHz. The bioheat and cell death models are used to analyse the heat distribution and cell response in the liver, respectively. A difficult task in microwave ablation technology is the computation of SAR. By fitting the curve to the experimental data, Gao et al. [9] arrived at the cubic polynomial for the SAR distribution. The SAR distribution of a 2.45 GHz MW antenna was presented at power levels of 40 W, 45 W, 50 W, 55 W, and 60 W. A novel wave propagation model in H-formulation or E-formulations is widely used to simulate SAR profiles [10–12]. A few researchers reduced the 3D problem to a 2D axisymmetric problem in $H_\Phi$–formulation with the help of axisymmetry in geometry [10, 11]. Tehrani et al. [13] considered 3D spherical geometry and solved the wave propagation model in E-formulation using the Nodal Finite Element Method (NFEM). Many studies have modeled the bio-heat equation to examine the heat distribution and temperature profile in live tissue [14–17]. Due to its simplicity, Penne's Bio-heat model for heat distribution in the liver has been frequently used in the computational field of ablation therapies [15]. Chen and Holmes arrived at the heat distribution equation by separating the domains of tissue and blood [17]. A local thermal non-equilibrium model that takes the convective term and blood velocity into account was obtained by Keangin and Rattanadecho [16]. In this study, Pennes' bio-heat equation has been used due to its simplicity in implementation. In this work, the three-dimensional domain in Cartesian coordinates is considered as the computational domain, and all three PDEs are solved in the same domain via FEniCS.

The obvious approach when using the Nodal Finite Element Method (NFEM) to handle vector-valued partial differential equation was to represent the vector field $\vec{E}$ as three related scalar fields, $E_x$, $E_y$, and $E_z$. To approximate the scalar fields, scalar elements were employed, however, this time each node contains three unknowns instead of one. When employing NFEM, we can obtain spurious solutions and convergence to solutions to problems that aren't related to our original problem. We can picture a basic cavity eigenvalue problem with a perfectly conducting wall as an illustration of the aforementioned [18, 19]. When solving Maxwell's equations, another significant disadvantage of NFEM is the inability to simulate field singularities at conducting corners and tips. At the conducting surface, the perfect electric boundary condition is $\vec{n} \times \vec{E} = 0$, but applying this condition at each node also enforces normal continuity, which is undesirable [20]. The Vector Finite Element Method (VFEM) has

been successfully implemented to alleviate the aforesaid issues in computational electromagnetic theory [21–23]. In comparison to NFEM, VFEM has a number of advantages in 3D computational electromagnetic field theory. The 3D domain was discretized using the tetrahedron, brick elements, and hexahedron elements. The continuity of only tangential components across interfaces between two neighboring finite elements is satisfied by vector elements. In comparison to the algorithm based on NFEM, the performance of the VFEM-based approach is higher and consumes fewer memory [24]. Rather than nodes, the degrees of freedom in VFEM were defined along edges as shown in Fig 1. The tetrahedral element has 12 degrees of freedom in NFEM, with three degrees of freedom at each node. In VFEM, the tetrahedral element has 6 degrees of freedom, with one degree of freedom at each edge. Therefore, in this paper, we clearly explained the implementation of VFEM for complex-valued wave propagation equations in the 3D domain with absorbing and perfect conducting boundaries.

The ablation zone volume and MWA efficiency are connected. The liver suffers cell damage when its temperature rises above 43°C [11] for an extended length of time. As a result, cell response models are function of temperature and heating time. The simplest models employ a single temperature threshold below which cells continue to operate normally and over which cells are instantly declared dead [25]. It is challenging to appropriately analyse the thermal damage since these models do not take the transition state between the alive and dead states into account. The Arrhenius model has been used in many research to examine how cells react to heat [26, 27]. These models have two main drawbacks: 1) The Arrhenius parameters are very sensitive to experimental values, 2) The cell survival rates at different temperatures reveal a clear discontinuity at temperatures around 43°C. In the literature [28], the Author proposed a mathematical cell death model for hyperthermia by introducing a vulnerable state between living and dead states. Moreover, this model exhibits the "shoulder region" found in experimental data [29].

Many factors, including frequency, input power, heating period, and antenna design, affect the ablation zone profile [10, 30–32]. The majority of the MWA system operated in the 915 MHz or 2.45 GHz frequency range. To perform MWA with less side effects and more efficacy in treating small-sized tumor, 18 GHz is the ideal frequency with low input power (1–3 W) [33]. In order to create a bigger ablation zone, Hines-Peralta [34] studies the impact of power

(a)                                        (b)

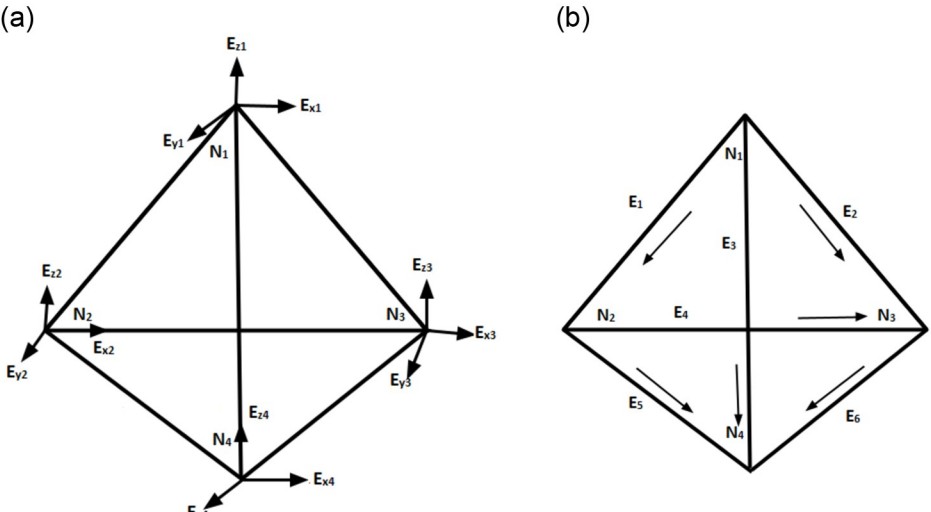

**Fig 1.** (a) Node-based tetrahedral element. (b) Edge-based tetrahedral element.

(50–150 W) and treatment duration in an ex vivo liver model. They found that ablation volume increased with heating duration and power input, with the biggest ablation volume occurring with the maximum heating time and power input. A few researchers used time-dependent input power to optimize the thermal damage for healthy cells [31, 35]. Mellal et al. [35] controlled the heat deposition and optimized the healthy cells damage by applying pulsating input power with a pulse on the duration of 2.5 s and a pulse off time of 5 s. Another crucial element in the microwave ablation procedure is the design of the antenna and the number of slots. Ibitoye et al. [32] examined the effectiveness of various MWA antenna proposals. According to their findings, the sleeve antenna generates the best sphericity ablation profile. Mellal et al. [35] designed a curved shape antenna to the direction of heat deposition. This antenna is helpful in treating small irregular tumors. In order to achieve a spherical SAR pattern, Etoz and Brace [36] built dual-slot antennas with an 8 mm gap between the slots. Marwa Selmi et al. [37] investigated temperature, the influence of the type of antenna on the temperature distribution in the breast tissue, the specific absorption rate, and the amount of necrotic tissue. Rattanadecho and Keangin [10] compared the effects on the SAR profile, temperature, and blood velocity profiles for single-slot and double-slot antennas. And it is found that the SAR, temperature, and blood velocity patterns when using a double slot antenna provide a wider region in the liver. Even though double-slot antenna have some advantages over single-slot antenna, we used single-slot antenna because of their simple geometry, low cost, ease of use, compact size, and great efficiency [10, 38]. Tissue undergoes mechanical deformation during the treatment due to water vaporization. Keangin et al. [39] included a tissue deformation model for the 2D axisymmetric liver geometry. The heat distribution for models with and without deformation has been studied. Considering mechanical deformation models for the 3D domain is a challenging task, and it is kept for future research.

The liver and tumor have different thermal and dielectric characteristics. As a result, the heat distribution in the tissue is influenced by the size and shape of the tumors. One of the difficulties in MWA is to destroy large tumors (diameter > 3 cm) with minimum damage to the healthy cells. Wang et al. [40] claimed that single-ablation (destroying the cells by generating the heat at one place throughout the treatment) with one antenna was suitable for cancers under 3 cm in diameter and that repeated overlapping ablation with one antenna is required for tumors between 3 and 5 cm in diameter. By increasing the input power and heating time, the single ablation approach can be used to treat large tumors. At the same time, high microwave power and heating time cause an excess of heat production in the liver, which kills the healthy cells. The dimensions of the ablation zone using a single ablation approach from a previously published article are mentioned in Table 1. Most of the antennas produce an oval-shaped ablation zone when a single ablation approach is used [32].

In cancer types like breast, lung, and liver, the tumor modules appear in a variety of sizes and shapes. The spherical shape is one of the common shapes, which covers most of the tumor shapes [12, 13, 41]. It is challenging to create a spherical-shaped ablation zone with a single ablation technique, as can be seen from the aspect ratio in Table 1. Laeseke et al. [42]

**Table 1. Dimensions of the ablation zone using a single ablation technique.**

| Research | Antenna | Power and Frequency | Diameter(D) and Length(L) of the ablation zone | Aspect ratio $= \frac{D}{L}$ |
|---|---|---|---|---|
| Tehrani et al. [13] | Single-slot | 50 W and 2.45 GHz | Diameter 29.2 mm, Length 46 mm | 0.64 |
| Tehrani et al. [13] | Double-slot | 50 W and 2.45 GHz | Diameter 32 mm and Length 72 mm | 0.44 |
| Ibitoye et al. [32] | Monopole | 40 W and 2.45 GHz | Diameter 32 mm, Length 51 mm | 0.62 |
| Ibitoye et al. [32] | Sleeved | 40 W and 2.45 GHz | Diameter 33 mm, Length 47 mm | 0.7 |

investigated the ablation zone by using multiple-antenna during the treatment. They inserted the three antennae in a triangular pattern at the same time with 30 W input power. Multiple-antenna technique helps to create a large ablation zone with a spherical shape. This technique is commercially high due to the use of multiple antennae in the treatment, and it is difficult to control the cell damage for normal-sized tumors. To kill the maximum volume of cancer cells with the minimum damage to healthy cells, heat has been produced at a few locations instead of overheating in one place. Today, the use of mathematical modeling and computational tools in treating cancer has become widespread and is progressing due to their cost-free and relatively low time, and simulates the complex system process without any influence of the test on the human body, animal species, or any biological tissues. This study mathematically simulated the heat distribution in the liver during the multi-ablation technique at different antenna locations and heating times.

According to the above-mentioned literature review, for the first time, the 3D vector finite element method was used to simulate an electric field from a wave equation, which is an important part of the field of computational microwave ablation. The main challenge associated with microwave ablation is to destroy the complete cancer cells with minimal damage to the healthy cells. From the above literature review, many researchers optimize thermal damage by setting time-dependent input power, an optimized frequency range, and directed antennas. Along with this, this study optimizes the thermal damage to healthy cells by producing heat at different places instead of overheating at one place. This study numerically simulated and analyzed heat distribution and ablation volume for the multi-ablation technique for large tumors by inserting the antenna sequentially into the liver to optimize thermal damage to healthy cells. A cell death model was used to analyze the shape and size of the ablation. Numerical results are validated with experimental results to validate the numerical methods, mathematical models, and parameters.

## Wave propagation model

The three-dimensional wave propagation model is obtained from the Maxwell equation based on the assumptions that tissue and antenna are linear, isotropic, and uniform mediums. The microwave power is propagated in the tissue. The complex-valued electric field intensity $\vec{E}(x, y, z)$ is calculated using the following equation [13, 43]:

$$\nabla \times \mu_r^{-1}(\nabla \times \vec{E}) - k_0^2 \varepsilon_R \vec{E} = 0 \tag{1}$$

where $\vec{E} = \vec{E}^{re} + j\vec{E}^{im}$ is the complex-valued electric field (V/m), $\mu_r$ is the relative permeability, $\varepsilon_R = \varepsilon_r - j\frac{\sigma}{\omega\varepsilon_0}$ is the complex relative permittivity, $\varepsilon_r$ is the relative permittivity, $\sigma$ is the electric conductivity of the tissue (S/m), $\varepsilon_0 = 8.8 \times 10^{-12}$ F/m is the vacuum relative permittivity, $k_0$ is the propagation constant in free space (m$^{-1}$), and dielectric properties of the antenna and tissue are listed in Table 2.

**Table 2. The dielectric properties of the antenna [13, 38].**

| Properties | $\epsilon_r$ | $\mu_r$ | $\sigma$ (S/m) |
| --- | --- | --- | --- |
| Dielectric | 2.03 | 1 | 0 |
| Catheter | 2.1 | 1 | 0 |
| Slot | 1 | 1 | 0 |
| Liver | 43.03 | 1 | 1.69 |
| Tumor | 48.03 | 1 | 2 |

The boundary conditions:

1. The conductor's boundary of the single-slot coaxial antenna is modeled as a perfect electric conductor boundary [44]:

$$\vec{n} \times \vec{E} = 0 \tag{2}$$

2. The first-order absorbing boundary condition is used at the input port to represent the boundary of the input port [22, 45].

$$\vec{n} \times (\nabla \times \vec{E}) = j\beta(\vec{E} - 2\vec{E}_{inc}) \tag{3}$$

where $\beta$ is the propagation constant, and $\vec{E}_{inc}$ is the incident electric field [46].

$$\vec{E}_{inc} = \vec{P} e^{j\beta \vec{k} \cdot \vec{r}}$$

where, $\vec{P}$ is the polarization vector, $\vec{k}$ is the propagation vector, and $\vec{r}$ is the position vector.

3. An absorbing boundary condition without incident field was imposed on the exterior boundary to truncate the computational domain [22].

$$\vec{n} \times \left( \nabla \times \vec{E} \right) + jk_0 \left( \vec{n} \times \left( \vec{n} \times \vec{E} \right) \right) = 0 \tag{4}$$

where $\vec{n}$ is a normal vector perpendicular to the external boundary.

The primary factor influencing temperature change in biological tissue is the presence of external heat which is generated by the electric field. Therefore, the external heat source term is given by [47]:

$$Q_e = \frac{1}{2} \sigma |\vec{E}|^2 \tag{5}$$

The SAR is defined as the energy absorbed by the unit mass of the tissue, which is obtained as follows [47]:

$$\text{SAR} = \frac{Q_e}{\rho} = \frac{1}{2\rho} \sigma |\vec{E}|^2 \tag{6}$$

## Heat distribution model

The following Pennes bio-heat equation describes the temperature distribution in the tissue during ablation therapy [15].

$$\rho C \frac{\partial T}{\partial t} = \nabla \cdot (k\nabla T) + Q_e + \rho_b \omega_b C_b (T_b - T) + Q_m \tag{7}$$

where $\rho$ is the tissue density (kg/m$^3$), $C$ is the effective specific heat capacity of the tissue (J/ kg·°C), $k$ is tissue thermal conductivity (W/m°·C), $T$ is the temperature (°C), $Q_e$ is the absorbed electromagnetic energy (W/m$^3$), $\omega_b$ is blood perfusion rate (1/s), $C_b$ is the specific heat capacity of blood (J/ kg·°C), $T_b$ is the blood temperature (°C), $Q_m = 33800$ W/m$^3$ is the metabolic heat generation [16, 48]. The parameter values used in the Eq (7) are listed in Table 3.

The amount of energy needed to increase the temperature of a unit mass of tissue by 1°C is known as the effective specific heat, which also includes the energy needed to evaporate tissue.

**Table 3. The thermal properties of a tissue, tumor, and blood [13, 38].**

| Parameters | Value | | |
|---|---|---|---|
| | Tissue | Tumor | Blood |
| Specific heat capacity, (J/kg·˚C) | 3540 | 3960 | 3600 |
| Baseline thermal conductivity, (W/m·˚C) | 0.497 | 0.57 | 0.45 |
| Density, (kg/m$^3$) | 1030 | 1040 | 1058 |

According to the literature [49], tissue water content plays a significant role in MWA when tissue reaches higher temperatures. Therefore, tissue effective specific heat and tissue water content were modeled as a function of temperature, as follows [49–51]:

$$C = C_t - \frac{\alpha}{\rho}\frac{dW(T)}{dT} \tag{8}$$

$$W(T) = \begin{cases} 0.778 - 0.778 \times \exp\left(\frac{T-106}{3.42}\right) & \text{for } T \leq 103^{\circ}C \\ 0.0289T^3 - 0.8924T^2 + 919.6T - 31573 & \text{for } 103^{\circ}C < T \leq 104^{\circ}c \\ 0.778 \times \exp\left(\frac{T-80}{34.37}\right) & \text{for } T > 104^{\circ}C \end{cases} \tag{9}$$

For more accuracy in the solution, the thermal conductivity of the tissue was considered as a function of temperature and represented as follows [30, 52].

$$k(T) = k_0 + \Delta k(T - T_0) \tag{10}$$

where $k_0$ is the baseline thermal conductivity (W/m·˚C), $\Delta k$ is the change in $k$ due to temperature, and $T_0$ is the reference temperature (˚C) at which $k_0$ has been measured.

Since microvasculature collapses when the temperature exceeds 60˚C, therefore, we assumed a constant blood perfusion rate value for each element until the temperature crossed 60˚C after which it was set to zero. The blood perfusion rate was modeled as a temperature-dependent piecewise model given as [25, 53, 54]:

$$\omega_b(T) = \begin{cases} \omega & \text{for } T < 60^{\circ}C \\ 0 & \text{for } T \geq 60^{\circ}C \end{cases} \tag{11}$$

where $\omega_0 = 0.0036$ s$^{-1}$ is the initial blood perfusion rate at $t = 0$.

The blood and initial temperatures are generally taken as body temperature 37˚C, and the boundary of the liver tissue is treated as a thermally insulated boundary, which is mathematically represented by [38]

$$\vec{n} \cdot \nabla T = 0 \tag{12}$$

## Cell death model

This study examines the thermal injury to cells during microwave ablation using the cell death model. This model uses a dynamic system of ordinary differential equations to show how the cell reacts to temperature [28]. The three states of a cell's response to temperature are the living state, the vulnerable state, and the dead state. Cells start to change from an alive to a dead state via the vulnerable state when the temperature rises.

$$A \underset{k_b}{\overset{k_f}{\rightleftharpoons}} V \overset{k_f}{\longrightarrow} D \tag{13}$$

$$A + V + D = 1 \tag{14}$$

Here, $A$ is the fraction of alive cells, $V$ is the fraction of the vulnerable cells that may move to either an alive state or a dead cells, $D$ is the fraction of the dead state, $k_f$ is the forward rate constant (1/s) and it describes the healthy cells leading to an injured state, and $k_b$ is the backward rate constant (1/s), which represents a self-healing process of injured cells. The fractions of alive and dead cells are defined as a dynamic system as follows [28, 55]:

$$\frac{dA}{dt} = -k_f A + k_b(1 - A - D) \tag{15}$$

$$\frac{dD}{dt} = k_f(1 - A - D) \tag{16}$$

$$k_f = k_{f0} e^{\frac{T}{T_k}}(1 - A) \tag{17}$$

where $k_{f0}$ is the scaling factor ($s^{-1}$) and $T_k$ is a parameter setting the rate of the exponential increase with temperature.

## Tissue localised contraction model

According to preliminary findings, tissue contraction was not constant across time or space; it was greatest in the near antenna and the start of the ablation [56]. Many research has lately taken into account how tissue shrinkage is affected by thermal ablation [56, 57]. Using theoretical and experimental models, Liu et al. [57] investigated how MWA affected tissue contraction. Their findings indicate that temperatures above $102.1°C$ caused more than 50% of volumetric contraction. Liu and Brace [56] used intraprocedural computed CT imaging to evaluate six ex vivo bovine liver samples during MWA. Even while contraction was greater at higher temperatures, time also played a big role. In order to simulate this, a weighted temperature-time (thermal dose) dependent model was created by comparing it with experimental data [57].

$$\text{TTI}(T,t) = \begin{cases} 0 & 0 \leq T \leq 12°C \\ 0.1573 \int_0^t (T - T_0)\, dt & 12°C < T \leq 44.1°C \\ 0.3011 \int_0^t (T - T_0)\, dt & 44.1°C < T \leq 102.1°C \\ 0.5416 \int_0^t (T - T_0)\, dt & 102.1°C < T \end{cases} \tag{18}$$

where $T_0$ is the initial temperature.

The weighted temperature-time integration (TTI) data at each location were fitted to the mean localised contraction data using the regression of a power function [57].

$$\text{LC}_r = 0.003 \cdot \text{TTI}^{0.4684} \tag{19}$$

To explain the tissue contraction at specific points within the ablation zone, a localised contraction model was developed.

## Implementation of 3D-VFEM to wave propagation equation

To implement NFEM into the wave propagation equation, one has to consider the wave propagation equation component-wise and define three degrees of freedom at each node. It results in high computation time and memory storage. In VFEM, degrees of freedom are defined

along the element's edges. The vector Nedelc basis functions approximate the electric field intensity over the computational domain.

**Weak formulation of propagation equation.** We can convert the strong form of the wave propagation equation to the weak formulation by choosing the test function from the following spaces.

$$H(curl; \Omega) = \{u \in [L_2(\Omega)]^3; \nabla \times u \in [L_2(\Omega)]^3\}$$
$$H_0(curl; \Omega) = \{u \in H(curl; \Omega); \vec{n} \times \vec{E} = 0 \quad \text{on} \quad \Gamma\}$$

where $\vec{n}$ is the unit normal to conductor boundary $\Gamma$ and $H(curl; \Omega)$ is Hilbert space with norm

$$\|u\|_{H(curl;\Omega)} = (\|u\|_2^2 + \|\nabla \times u\|_2^2)^{\frac{1}{2}}$$

Since the wave propagation equation is a complex-valued differential equation, the original equation is split into real and imaginary parts. After that, one can construct the real-valued variational form to determine the $\vec{E}^{re}(x, y, z), \vec{E}^{im}(x, y, z) \in H(curl; \Omega)$ such that

$$
\begin{aligned}
&\mu_r^{-1} \int_\Omega \left( \nabla \times \vec{E}^{re} \right) \cdot \left( \nabla \times \vec{W} \right) d\Omega - k_0^2 \varepsilon_R^{re} \int_\Omega \vec{E}^{re} \cdot \vec{W} d\Omega + k_0^2 \varepsilon_R^{im} \int_\Omega \vec{E}^{im} \cdot \vec{W} d\Omega \\
&-\beta \mu_r^{-1} \int_{S_p} \vec{E}_{x-y}^{im} \cdot \vec{W} dS + k_0 \mu_r^{-1} \int_{S_l} \left( \vec{n} \times \vec{E}^{im} \right) \cdot \left( \vec{W} \times \vec{n} \right) dS = -2\beta \mu_r^{-1} \int_{S_p} \vec{E}_{inc}^{im} \cdot \vec{W} dS
\end{aligned}
\tag{20}
$$

$$
\begin{aligned}
&\mu_r^{-1} \int_\Omega \left( \nabla \times \vec{E}^{im} \right) \cdot \left( \nabla \times \vec{W} \right) d\Omega - k_0^2 \varepsilon_R^{im} \int_\Omega \vec{E}^{re} \cdot \vec{W} d\Omega - k_0^2 \varepsilon_R^{re} \int_\Omega \vec{E}^{im} \cdot \vec{W} d\Omega \\
&+\beta \mu_r^{-1} \int_{S_p} \vec{E}_{x-y}^{re} \cdot \vec{W} dS - k_0 \mu_r^{-1} \int_{S_l} \left( \vec{n} \times \vec{E}^{re} \right) \cdot \left( \vec{W} \times \vec{n} \right) = +2\beta \mu_r^{-1} \int_{S_p} \vec{E}_{inc}^{re} \cdot \vec{W} dS
\end{aligned}
\tag{21}
$$

for all $\vec{W} \in H_0(curl; \Omega)$. where $S_p$ is the input port boundary and $S_l$ is the liver boundary.

**Construction of edge tetrahedral elements for 3D domain.** The following provides a general definition of finite elements on arbitrary tetrahedral. A finite element $(K, P, A)$ consists of

1. $K$, a tetrahedral domain

2. $P$, a polynomial space defined on $K$ having basis function $\{\vec{N}_1, \vec{N}_2, ..., \vec{N}_n\}$

3. $A$, a set of linear functional defined on $P$ having a basis functions $\{\alpha_1, \alpha_2, ..., \alpha_n\}$

$$\alpha_i(v) = \int_{e_i} (v \cdot t_i) ds \qquad v \in P$$

where $t_i$ is unit tangent vector along edge $e_i$ of $K$.

The polynomial space is defined by considering the reference elements $(K_0, P_0, A_0)$, as shown in Fig 2. Each of the tetrahedral can be mapped to a reference element $K_0 = \{0 \leq \xi, v, \eta \leq 1\}$. The reference 3D tetrahedral element is illustrated in Fig 2. The basis functions for

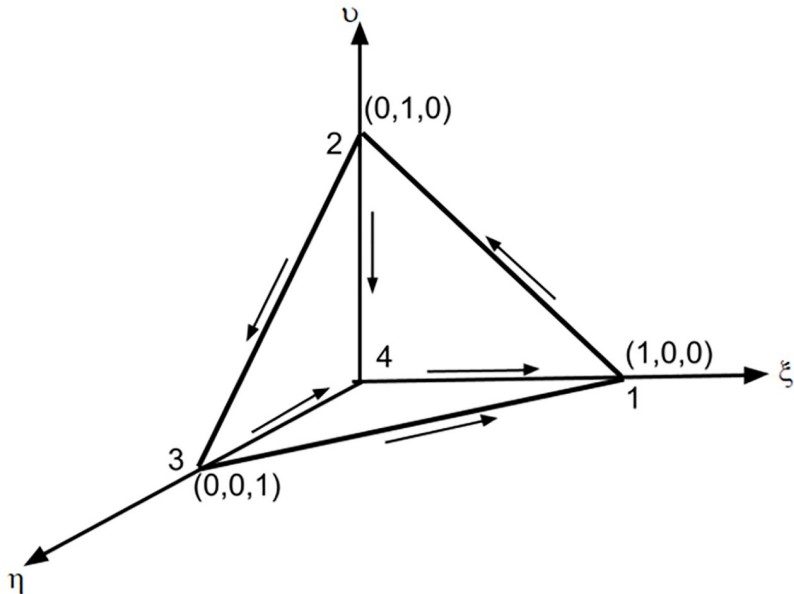

**Fig 2. Reference tetrahedral element.**

reference space $P_0$ are given as

$$
\begin{aligned}
\vec{N}_{41}^0 &= (1 - v - \eta, \xi, \xi), \quad \vec{N}_{12}^0 = (-v, \xi, 0), \\
\vec{N}_{24}^0 &= (-v, -1 + \xi + v, -v), \quad \vec{N}_{23}^0 = (0, -\eta, v), \\
\vec{N}_{34}^0 &= (-\eta, -\eta, -1 + \xi + v), \quad \vec{N}_{31}^0 = (\eta, 0, -\xi).
\end{aligned}
$$

The above polynomial basis function defined on the reference element must be transformed to the arbitrary tetrahedral $K$ by using affine element map $(x, y, z) = B_K(\xi, v, \eta) + b_K$.

**Galerkin approach for wave propagation equation.** It is straightforward to discretize the variational form using the vector basis function since the unknown variables are vector-valued functions. The Galerkin method expresses unknown variables as a linear combination of vector basis functions

$$
\vec{E}^{\mathrm{re}}(x, y, z) = \sum_{l=1}^{n_e} \vec{N}_l(x, y, z) E_l^{\mathrm{re}}, \quad \vec{E}^{\mathrm{im}}(x, y, z) = \sum_{l=1}^{n_e} \vec{N}_l(x, y, z) E_l^{\mathrm{im}}, \tag{22}
$$

where $n_e$ is the number of the edges in the element, $E_l^{\mathrm{re}}$ and $E_l^{\mathrm{im}}$ are tangential components of $\vec{E}^{\mathrm{re}}$ and $\vec{E}^{\mathrm{im}}$ respectively along the edge $l$, and $\vec{N}_l$ is the vector basis function defined along the edge $l$.

By choosing $\vec{W} \in \{\vec{N}_1, \vec{N}_1, ..., \vec{N}_{n_e}\}$, one can transform Eqs (20) and (21) to system of algebraic equations

$$
\begin{bmatrix} A^{\mathrm{re}} & A^{\mathrm{im}} \\ -A^{\mathrm{im}} & A^{\mathrm{re}} \end{bmatrix} \begin{bmatrix} E^{\mathrm{re}} \\ E^{\mathrm{im}} \end{bmatrix} = \begin{bmatrix} b^{\mathrm{re}} \\ b^{\mathrm{im}} \end{bmatrix} \tag{23}
$$

where

$$a_{kl}^{\text{re}} = \mu_r^{-1} \int_{\Omega_e} \left( \nabla \times \vec{N}_k \right) \cdot \left( \nabla \times \vec{N}_l \right) d\Omega - k_0^2 \varepsilon_R^{\text{re}} \int_{\Omega_e} \vec{N}_k \cdot \vec{N}_l d\Omega,$$

$$a_{kl}^{\text{im}} = k_0^2 \varepsilon_R^{\text{im}} \int_{\Omega_e} \vec{N}_k \cdot \vec{N}_l d\Omega - \beta \mu_r^{-1} \int_{S_p} \vec{N}_k \cdot \vec{N}_l dS - k_0 \mu_r^{-1} \int_{S_l} \left( \vec{n} \times \vec{N}_k \right) \cdot \left( \vec{N}_l \times \vec{n} \right) dS,$$

$$b_k^{\text{re}} = -2\beta \mu_r^{-1} \int_{S_p} \vec{N}_k \cdot E_{\text{inc}}^{im} dS,$$

$$b_k^{\text{im}} = 2\beta \mu_r^{-1} \int_{S_p} \vec{N}_k \cdot E_{\text{inc}}^{re} dS,$$

After getting the stiffness matrix for each element, the stiffness matrix was assembled using FEniCS, [58] and MUltifrontal Massively Parallel sparse direct Solver(MUMPS) was used to solve the system of equations.

The implementation of the finite element method and the finite difference method for the bioheat equation and cell death model is explained in S1 and S2 Appendices in S1 File, respectively.

## Locations of the antenna in the liver

The effectiveness of ablation techniques is determined by the ablation zone created during the treatment. For large tumors (diameter > 3 cm), using a single antenna at the center of the tumor is inefficient to kill the tumor in a short period of time [40]. Therefore, in this research work, we simulated and analyzed the results by introducing a single-slot antenna sequentially to the tissue at different positions. The antenna's position is chosen based on the literature [59]. The overall lesion form became less rounded, and clefts began to emerge when probe separations were higher than 17 mm. As a result, 14 mm is used as the distance between the two positions in order to avoid leaving non-ablated tissue between the two positions. The antenna's tip is sequentially positioned in the following locations: $P_1 = (-7 \text{ mm}, 7 \text{ mm}, 10 \text{ mm})$, $P_2 = (7 \text{ mm}, 7 \text{ mm}, 10 \text{ mm})$, $P_3 = (-7 \text{ mm}, -7 \text{ mm}, 10 \text{ mm})$, and $P_4 = (7 \text{ mm}, -7 \text{ mm}, 10 \text{ mm})$. Depending on the size of the tumor, the heating time can be adjusted. The algorithm for changing the location of the antenna is given as follows:

$$\mathcal{L}(\vec{X}, t) = \begin{cases} P_1 & \text{if} \quad 0 \leq t \leq t_1 \\ P_2 & \text{if} \quad t_1 < t \leq t_2 \\ P_3 & \text{if} \quad t_2 < t \leq t_3 \\ P_4 & \text{if} \quad t_3 < t \leq t_4 \end{cases} \tag{24}$$

where $\mathcal{L}(\vec{X}, t)$ is the location of the antenna at particular time, $[0, t_1]$, $(t_1, t_2]$, $(t_2, t_3]$, and $(t_3, t_4]$ are the heating time interval at the positions $P_1$, $P_2$, $P_3$, and $P_4$, respectively.

## Numerical simulation

The governing equations mentioned in the previous section are solved by using numerical techniques instead of analytical methods due to the complex geometry and non-linearity in the equation. The VFEM and FEM are used to solve the electromagnetic equation and bioheat equation, respectively. The time derivative in the bioheat equation is discretized using the unconditionally stable Euler backward scheme. In order to determine how much heat is distributed throughout the liver, the simulation first solves Maxwell's equations with boundary

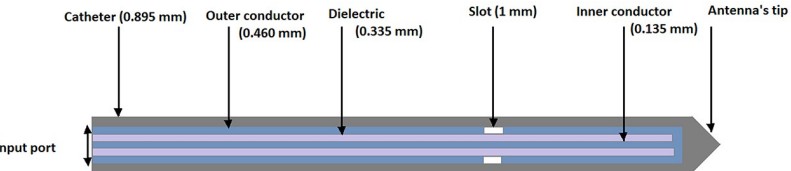

**Fig 3. The microwave heat deposition (W/m³) within the liver tissue at position $P_1$ using input power 50 W and frequency 2.45 GHz.**

conditions. The tumor is modeled as a sphere with a radius of 2 cm, whereas the liver is modeled as a cylinder with a height of 8 cm and a radius of 4 cm. The geometry and dimensions of the antenna are depicted in Fig 3. The domain is discretized using tetrahedral and triangular elements in Gmsh. A fine mesh is created inside the antenna and around the antenna's tip in order to obtain an accurate approximation. The FEniCS is used to solve the system of partial differential equations with the help of Nedelec and Legrange elements. A core i5 intel 10<sup>th</sup> generation CPU with 8GB memory RAM was used for simulation.

## Validation

The computational results are compared with the experimental data [50] and previously published results [13] in order to validate the parameters, models, and numerical methodologies of the present work. The microwave energy with an input power of 75 W and a frequency of 2.45 GHz is applied to the liver tissue through the single-slot coaxial antenna. Two locations in the liver, 4.5 mm and 7 mm radially away from the slot were used to measure the temperature distribution within the liver. At positions, 4.5 mm and 7 mm, the root mean square error between the numerical result and experimental result is 7.33˚C and 4.8˚C, respectively. The temperature distribution in the liver obtained by the present study was compared with the experimental results reported by Yang et al. [50], as shown in Fig 4.

Fig 5a shows the cross-sectional view of the ablation zone. The ablation zone (D+V) is obtained for an input power of 50 W and frequency of 2.45 GHz using the three-state cell

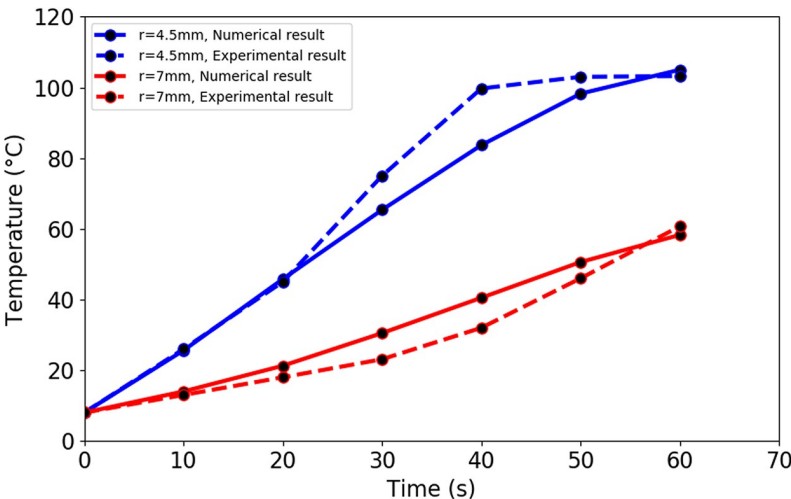

**Fig 4. Validation of temperature distribution obtained by present study against the experimental result.**

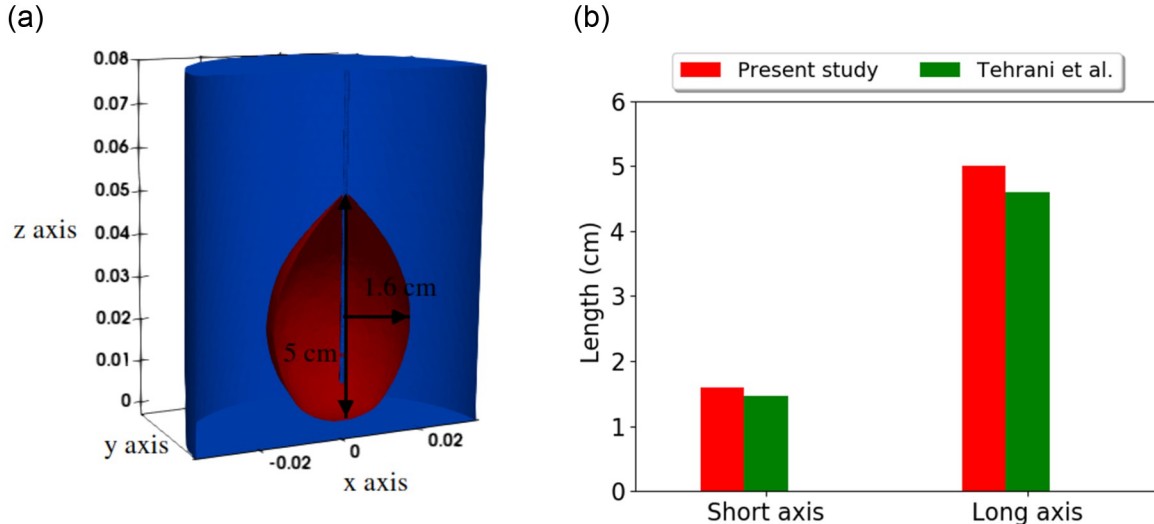

**Fig 5.** (a) Ablation zone obtained from cell death model using single antenna at input power 50 W and frequency of 2.45 GHz. (b) Comparison of ablation zone obtained from the present study and previously published results.

death model. The input power, frequency, and antenna geometry are taken the same as in the literature [13] to compare the cell death model results. The ablation zone dimensions obtained in the present study are in good agreement with the previously published results. Fig 5b compares the short and long axis of the ablation zone in the numerical results of the present study and results reported by Tehrani et. al. [13]. In comparison to previously published results for the short axis and long axis, the results showed differences of 0.15 cm and 0.4 cm, respectively.

## Results and discussion

In this section, pro/cons of single-ablation and multi-ablation techniques are discussed, numerical results are validated with experimental results to validate the numerical technique, parameters, and models. For an input power of 50 W and a frequency of 2.45 GHz, the heat deposition, temperature distribution, and ablation zones are simulated and analyzed at each position. The antenna is placed inside the tumor based on the location function $\mathcal{L}(\vec{X}, t)$. In the simulation, the rate of blood perfusion, thermal conductivity, and water vaporization are all treated as piecewise temperature-dependent functions.

To treat the majority of regular tumors with a diameter less than 3 cm, a single-ablation technique was usually sufficient, but for those highly irregular and large tumors, a single-ablation technique was not a good choice [40]. One can use this technique for large tumors by increasing input power and heating time. As we know from the literature [31], increasing heating time and input power can create an excess of heat in the liver, which can damage healthy cells. The spherical shape covers most of the tumor shapes; therefore, creating an ablation zone in a spherical shape is a necessary step in the ablation technique [12, 13]. For most of the antenna geometry, the single-ablation technique creates an oval-shaped ablation zone. Handling single-ablative technique is easy compared to multi-ablative technique clinically. Since finding locations for antennas is not difficult, patients can recover quickly with a single insertion in the body.

The multi-ablation technique is useful for large regular/irregular tumor with proper antenna locations and heating time. It helps us create a rounded ablation zone and control

healthy cell damage. Finding the location of the antenna, the heating time at each location, and the antenna separation distance is a difficult challenge in the multi-ablation technique. One has to take care while choosing the location and antenna separation distance for a given tumor size. The relationship between the separation distance, heating time, and tumor size for the multi-ablation technique is left for future investigation. Handling the multi-ablation technique is difficult compared to the single-ablation technique since new location has to be found after every ablation.

The microwave energy at a frequency of 2.45 GHz is transferred inside of the tumor via a single-slot coaxial antenna to produce heat, which kills the tumor. Polar molecules and ions in the tissue are constantly realigning with the alternating field, which causes heat to be produced [60]. The antenna is placed in the tumor at position $P_1$ in the beginning, and then it is placed at positions $P_2$, $P_3$, and $P_4$ in order. The antenna is placed at each location for 3 min to produce a sufficient amount of heat in the tissue. At each position, near the slot and tip of the single-slot microwave antenna, more heat from electromagnetic waves is deposited. As a result, there are heat deposits in the form of an oval shape around the antenna's slot and tip. One of the most important factors in temperature distribution in biological tissue is the SAR, which is defined as the heat absorbed by the tissue. The SAR value reaches its peak close to the slot and begins to decrease as it approaches the liver boundary. Fig 6 illustrates the heat deposition in the liver due to electromagnetic waves at position $P_1$. A few researchers simulated heat deposition in the liver without considering the tumor in the computational domain [11, 35, 61, 62]. In this work, results are simulated in the presence of a tumor with a diameter of 4 cm. To avoid the impact of the boundary conditions, the volume of the liver is considered to be five times greater than the tumor in the simulations. Dielectric properties in the tumor and liver are different because of structural differences [13].

The mechanism of cellular necrosis in the microwave ablation modality depends on both temperature distribution and heating time. With a 2–3 min exposure, temperatures exceeding 50˚C produce irreversible protein denaturation. Nuclear denaturation and other mechanisms have been related to nearly instantaneous cell death at temperatures greater than 60˚C [60]. Water vapourization and structural changes in the tissue occur at temperatures close to 100˚C

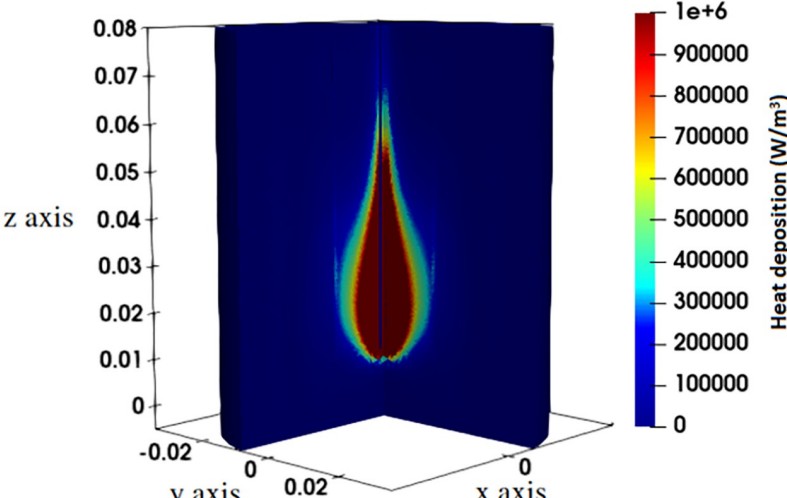

**Fig 6.** Temperature distribution using input power 50 W and frequency of 2.45 GHz in the liver along the line parallel to the antenna at (a) position $P_1$ (b) position $P_2$ (c) position $P_3$ (d) position $P_4$.

[49]. The MWA's goal is to create and maintain a zone of temperature exceeding 50˚C for 2–3 min or above 60˚C for a few seconds. Therefore, the antenna is placed according to location function $\mathscr{L}(\vec{X}, t)$ with $t_1 = 180$ sec, $t_2 = 360$ sec, $t_3 = 540$ sec, and $t_4 = 720$ sec instead of keeping the antenna at a single location for a long time. Heat conduction causes the temperature to increase as a result of the microwave energy being absorbed by the liver tissue and converted to thermal energy. The temperature profile in the liver was calculated numerically from the bioheat equation using FEM via FEniCs. The maximum temperature is attained near the slot and tip of the antenna since microwave heat deposition is near the antenna's slot and tip. The temperature decreases as it moves towards the liver boundary and reaches approximately body temperature. The temperature is monitored in the liver along the line parallel (5 mm away from the antenna) to positions $P_1$, $P_2$, $P_3$, and $P_4$ for time intervals $[0, t_1]$, $(t_1, t_2)$, $(t_2, t_3)$, and $(t_3, t_4)$, respectively, as shown in Fig 7. It's crucial to pay attention to whether the area around the antenna exceeds 60˚C for a few seconds. As a result, the input power, frequency, and antenna

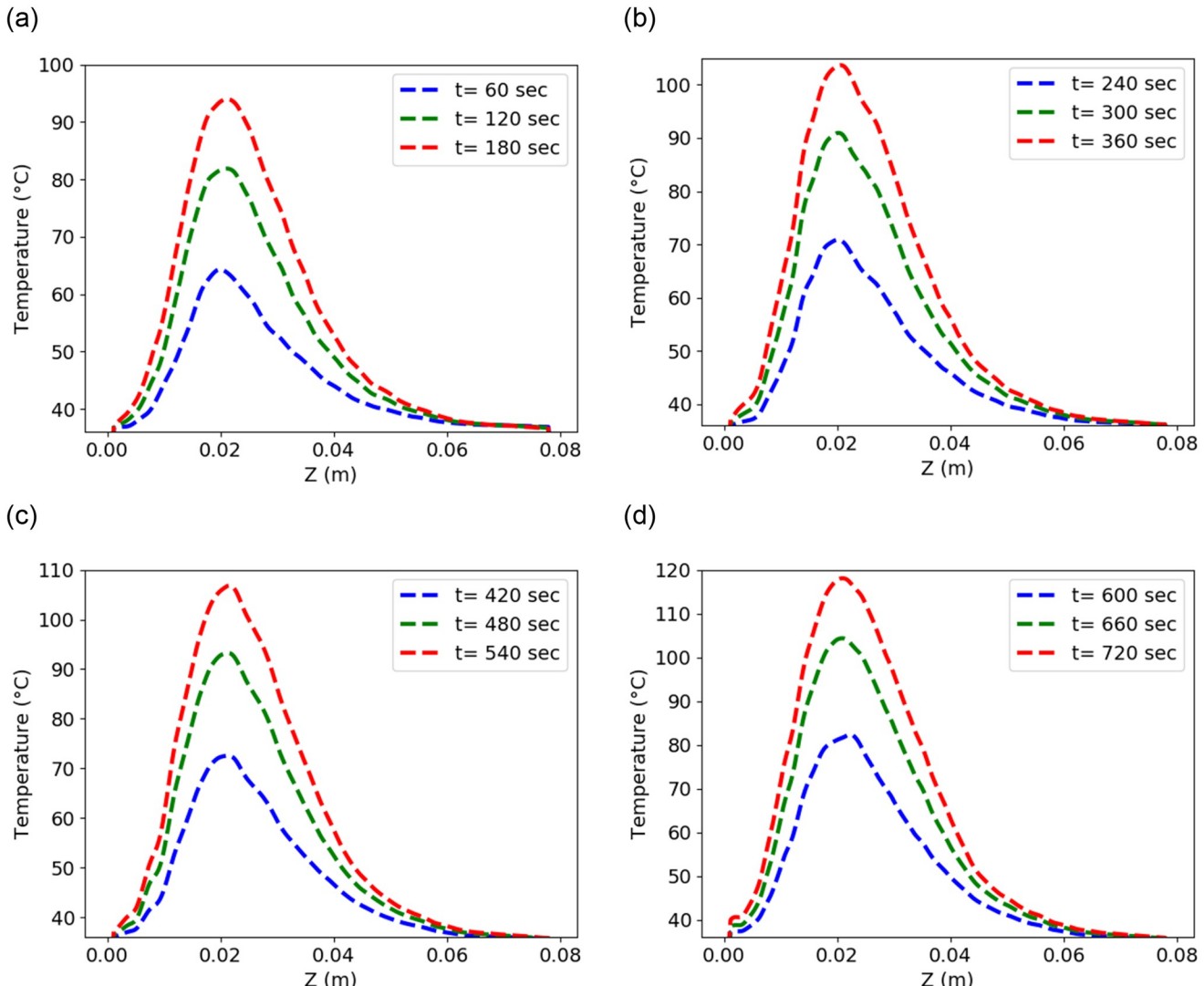

**Fig 7. Temperature profile in the liver at input power 50 W and frequency of 2.45 GHz during the treatment.**

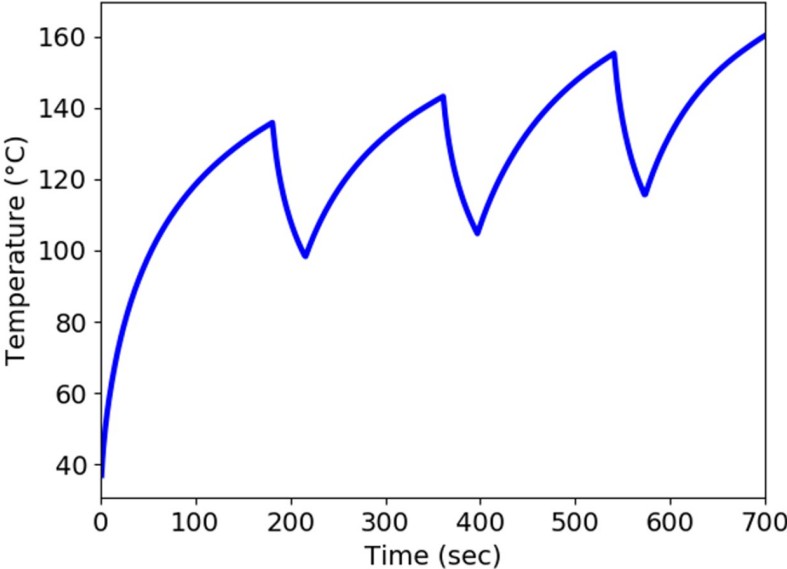

**Fig 8. Localized contraction at microwave power 50 W at 3 mm and 5 mm away from the position $P_1$, $P_2$, $P_3$, and $P_4$ for time interval [0, 180], [180, 360], [360, 540], and [540, 720], respectively.**

used in this research have the potential to destroy the cancer cells surrounding the antenna. The temperature distribution in the liver is directly dependent on the input power. One should use the input power according to the tumor size to avoid thermal damage to healthy cells [38]. Throughout the length of the treatment, the temperature profile in the liver moves in a zig-zag pattern. When the antenna is moved to a different place, the temperature begins to drop until equilibrium is reached, as shown in Fig 8.

The outcome of treatment is significantly impacted by tissue contraction during high-temperature thermal ablation. Recent research has taken into account how heat ablation affects tissue contraction [56, 57]. The mean localised contraction was monitored at different positions near the antenna at different time intervals. The contraction was greater at the higher temperatures; therefore, localised contractions are measured at positions parallel to the slot of the antenna. Fig 9 show the mean localised contraction for time periods [0, 180], [180, 360], [360,

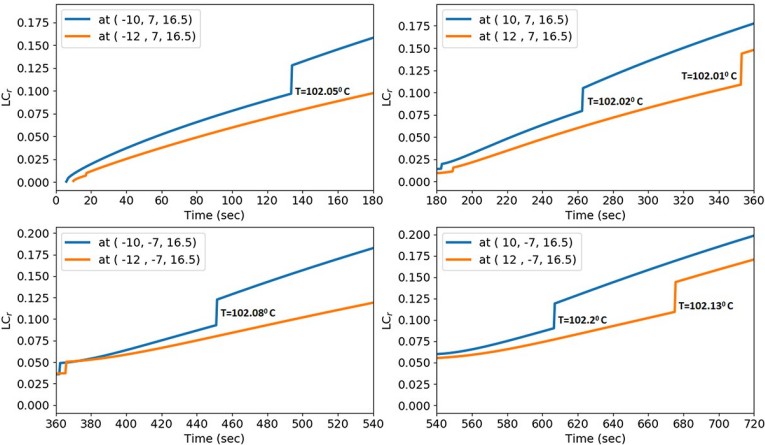

**Fig 9.** Ablation zone at (a) t = 180 sec (b) t = 360 sec (c) t = 540 sec (d) 720 sec.

540], and [540, 720] at the points 3 mm and 5 mm away from the positions $P_1$, $P_2$, $P_3$, and $P_4$, respectively. The tissue contraction decreases as it moves away from the antenna. At temperatures above 102˚C, the localised contraction model yields increased tissue contraction, which is consistent with the experiment results mentioned by Liu and Brace [57]. Localized contraction describes the local contribution to total tissue contraction. It showed that tissue contracted approximately by 15–20% and 7.5–12.5% near points 3 mm and 5 mm away from the antenna, respectively. According to previous investigations [56, 57, 63], which demonstrated that tissue contraction increased close to the antenna in a higher-temperature ablation zone with considerable water vaporization, the temperature and time-dependent contraction model indicated greater tissue contraction occurred at temperatures over 102˚C. This study focused only on local tissue contraction near the antenna at different locations. Liu and Brace [57] introduce heat-induced volume contraction as an energy term in the bio-heat equation and estimate the temperature. The effectiveness of any ablative technique is measured by the ablation zone created during the treatment. The ablation zone is numerically calculated using the three-state cell death model via the finite difference method. To avoid non-ablated tissue between the two locations, a 14 mm gap is used as the spacing between them. To provide a uniform heat distribution and a sphere-shaped ablation zone at the end of the treatment, the antenna positions are selected in a square arrangement. Fig 10 shows the top view of the ablation zone at a heating time, $t$ = 180, 360, 540, and 720 sec and antenna locations $P_1$, $P_2$, $P_3$, and $P_4$ respectively. The volume of the ablation zone increases with heating time. One of the aims of this work is to introduce the antenna in a sequential configuration for large tumors. Therefore, the tumor is taken as a sphere with a diameter of 4 cm, which has a volume of 33.5 cm$^3$. The ablation zone is simulated for the following cases: 1) Antennas are inserted at four locations sequentially, with 3 and 4 minutes of heating time at each location, 2) Antennas are inserted at three locations sequentially in a triangle pattern, with 5 minutes of heating time at each location, and 3) Antenna inserted at a single location, with heating time 20 min. The volume of the ablation zone, thermal damage, and remaining tumor cells are mentioned in Table 4. When the antenna was placed for 4 min at each position, 95.5% of the tumor cells and 1.8 cm$^3$ of healthy cells were killed. 70% of cancer cells were killed when antennas were inserted at three locations sequentially with a heating time of 5 minutes at each location. The single ablation technique took 20 minutes to kill 95.97% of the tumor, and 7.76 cm$^3$ of healthy cells were killed. The multi-ablation technique optimizes the thermal damage compared to the single-ablation technique for large tumors. Increasing heating time and input power can destroy complete tumor cells. Implementing pulsating power in the treatment instead of constant input power optimizes healthy cell damage [31, 35]. The volume of the ablation zone and thermal damage depend on the number of locations, the separation distance between the locations, and the heating time at each location. One can optimize the efficiency of the mult-ablation technique by optimizing the above factors.

## Conclusion

It is challenging to come up with an analytical solution for governing PDEs because of the geometry and complexity of the problem. Thus, the wave propagation, bio-heat, and cell death models are solved using numerical techniques. This study explains the implementation of the 3D vector finite element method for the wave propagation model to simulate the electric field intensity and heat deposition in the liver during the treatment. The 3D tetrahedral elements are used to discretize the domain via the open software Gmsh and Nedelec basis and test functions are chosen from $H_0(curl;\Omega)$ space to discretize the wave propagation equation. This implementation can solve many problems involving electromagnetic propagation with perfect

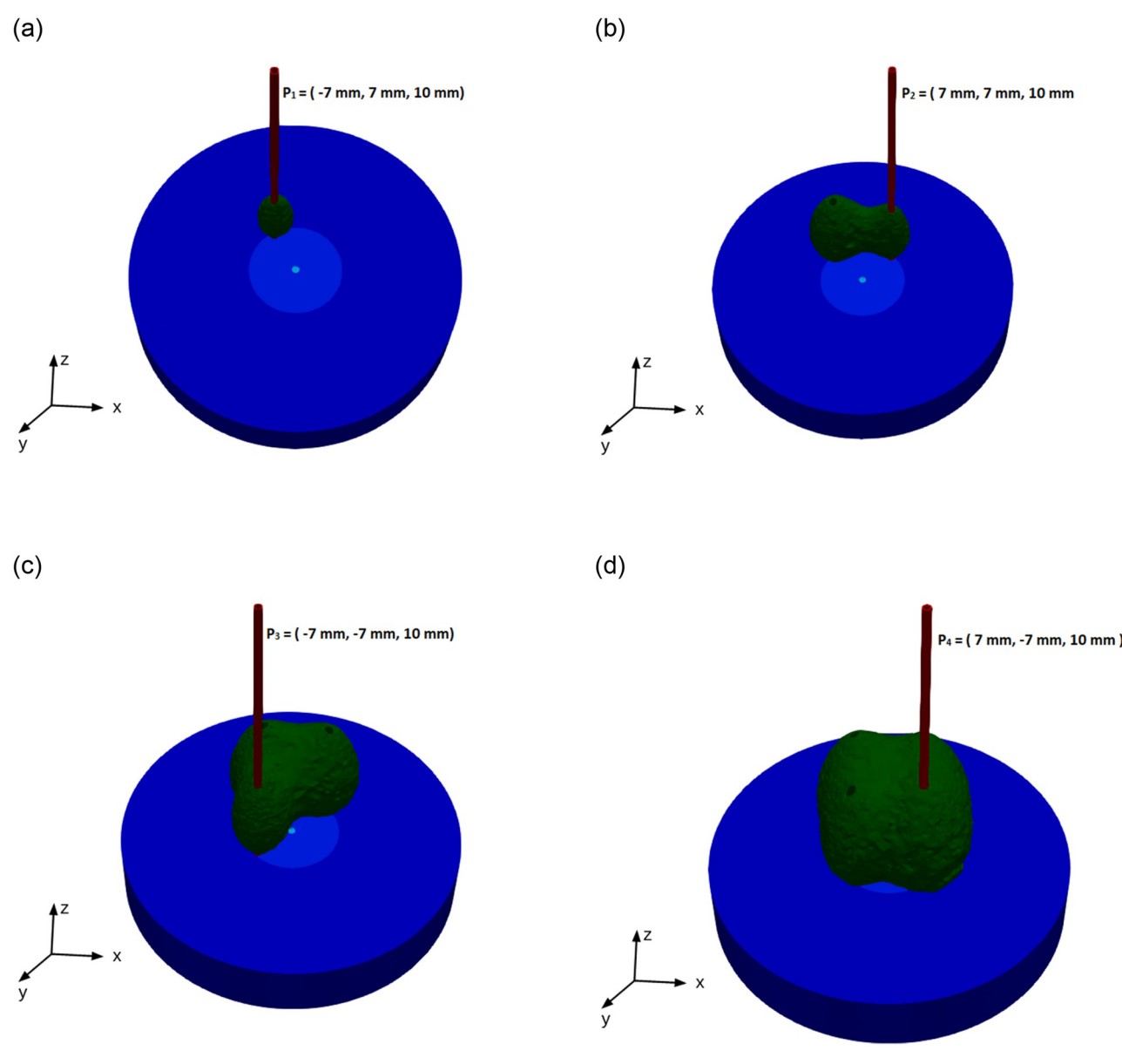

**Fig 10.**

**Table 4. The thermal damage in the tissue after the treatment.**

| Time | Ablation volume (cm$^3$) | Healthy cells damage after treatment (cm$^3$) | Remaining tumor cells after treatment (cm$^3$) |
|---|---|---|---|
| Multi-ablation (four locations), 3 min at each position | 20.9 | 0 | 12.6 |
| Multi-ablation (four locations), 4 min at each position | 33.7 | 1.8 | 1.55 |
| Multi-ablation (three locations), 5 min at each position | 26.002 | 3 | 10.5 |
| Single-ablation (one location), 20 min at one position | 40.26 | 7.76 | 1.7 |

conductors and absorbing boundaries. The bio-heat equation and cell death models are solved using finite element analysis and the finite difference method, respectively, via FeniCS.

Computationally, heat distribution and ablation zone are analyzed by generating heat by placing single-slot coaxial antenna at positions $P_1$, $P_2$, $P_3$, and $P_4$ sequentially for different heating times. We observed that the temperature profile during the treatment followed a zig-zag pattern due to a sudden change in the antenna's location in the domain, and the shape of the ablation zone formed a rounded shape after the treatment due to uniform heat distribution. Therefore, this technique controls the excess heat production in the liver during the treatment.

During the treatment, the tissue contraction is analysed using the weighted temperature-time-dependent and power function models. The local tissue contraction was observed at arbitrary points 3 mm and 5 mm away from the antenna slot radially, and it was more pronounced at temperatures greater than 102˚C. The size and shape of the ablation zone are obtained by solving the cell death model. The volume of the ablation zone was investigated for various settings, 95.5% of the cancer cells were killed with minimum thermal damage when antennas were inserted sequentially at four locations. One has to choose the heating time based on the size of the tumor. The relation between the antenna separation distance, heating time, and tumor volume for the multi-ablation technique is left for future investigation.

## Supporting information

**S1 Data.**
(ZIP)

**S1 File.**
(PDF)

## Author Contributions

**Conceptualization:** Gangadhara Boregowda, Panchatcharam Mariappan.

**Methodology:** Gangadhara Boregowda, Panchatcharam Mariappan.

**Software:** Gangadhara Boregowda.

**Supervision:** Panchatcharam Mariappan.

**Validation:** Gangadhara Boregowda.

**Writing – original draft:** Gangadhara Boregowda.

**Writing – review & editing:** Panchatcharam Mariappan.

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
