## [Decision Letter · Decision Letter 0]

14 Feb 2023

PONE-D-22-351343D Modeling of  Vector/Edge Finite Element Method for Multi-ablation  Technique for Large Tumor- Computational ApproachPLOS ONE

Dear Dr. boregwoda,

Thank you for submitting your manuscript to PLOS ONE. After careful consideration, we feel that it has merit but does not fully meet PLOS ONE’s publication criteria as it currently stands. Therefore, we invite you to submit a revised version of the manuscript that addresses the points raised during the review process.

We look forward to receiving your revised manuscript.

Kind regards,

Cihun-Siyong Alex Gong, Ph.D.

Academic Editor

PLOS ONE

Journal Requirements:

4. PLOS requires an ORCID iD for the corresponding author in Editorial Manager on papers submitted after December 6th, 2016. Please ensure that you have an ORCID iD and that it is validated in Editorial Manager. To do this, go to ‘Update my Information’ (in the upper left-hand corner of the main menu), and click on the Fetch/Validate link next to the ORCID field. This will take you to the ORCID site and allow you to create a new iD or authenticate a pre-existing iD in Editorial Manager. Please see the following video for instructions on linking an ORCID iD to your Editorial Manager account: https://www.youtube.com/watch?v=_xcclfuvtxQ.

5. Please ensure that you refer to Figure 2 in your text as, if accepted, production will need this reference to link the reader to the figure.

6. Please include a separate caption for each figure in your manuscript.

Additional Editor Comments:

Please revise your manuscript thoroughly according to ALL the referees's comments before resubmitting it

Reviewers' comments:

Reviewer's Responses to Questions

**Comments to the Author**

1. Is the manuscript technically sound, and do the data support the conclusions?

Reviewer #1: Yes

Reviewer #2: Yes

Reviewer #3: Yes

2. Has the statistical analysis been performed appropriately and rigorously? 

Reviewer #1: N/A

Reviewer #2: N/A

Reviewer #3: N/A

3. Have the authors made all data underlying the findings in their manuscript fully available?

Reviewer #1: No

Reviewer #2: No

Reviewer #3: Yes

4. Is the manuscript presented in an intelligible fashion and written in standard English?

Reviewer #1: No

Reviewer #2: Yes

Reviewer #3: Yes

5. Review Comments to the Author

Reviewer #1: This manuscript entitled "3D Modeling of Vector/Edge Finite Element Method for Multi-ablation Technique for Large Tumor- Computational Approach" implemented the 3D-vector/edge finite element method to simulate the specific absorption rate of microwave ablation (MWA). There are some major points to resolve prior to publication. Therefore, a major revision is recommended. The following points may help during the revision process.

1)The novelty of the paper should be clarified in the abstract and conclusion.

2)Authors should include some other qualitative and quantitative results in the abstract.

3)The overview in the introduction is not complete. It is necessary to review the literature and supplement some relevant literature appropriately. An updated literature review should be conducted based on recent novel studies. It can be helpful to present the state-of-the-art and knowledge gaps in the research.

4)Some of your equations don’t have a prestigious reference.

5)Validation should be performed in a specific new section.

6)Conclusions should be stronger and arranged into a few substantive and interconnected paragraphs instead of a short lump of text. The language is deficient in places.

7)There are some grammatical errors in the manuscript. Be sure to address them.

Reviewer #2: Reviewer report of the manuscript “3D Modeling of Vector/Edge Finite Element Method for Multi-ablation Technique for Large Tumor- Computational Approach” by G. Boregowda and P. Mariappan

In this contribution, the authors present some numerical simulations about microwave ablation applied on liver tissue by using Pennes' equation coupled with the wave propagation equation to predict the electromagnetic field. The main novelty here is that the authors use a four-antennas procedure to improve the extension of the ablated zone; furthermore, the authors also implement the 3D-vector/edge finite element to compute the SAR here. Results have been validated with comparisons with experimental data from the literature. From the simulations, results are shown in terms of temperature distribution and tissue contraction, showing the potential of the procedure here shown in terms of induced thermal damage for the tumor tissue to be treated.

The reviewer thinks that this is a well-presented paper that focuses about a very recent topic, that is MW ablation. It is widely known that optimizing MW ablation predictions to optimize medical procedures still remains a challenging task, and the results here shown are something that might be helpful for future improvements of this technique. Therefore, it is suggested to consider this contribution for publication if the following answers are properly addressed.

-Within the abstract, it is suggested to include more outcomes about results here achieved in terms of ablated zones, as done by the authors within the conclusions

-Please define the aspect ratio when introducing Table 1

-Please report references for the antenna geometry when introducing governing equations. Many geometries have been proposed through the years [1-3], so why did the authors decided to use exactly this antenna? Please note that ablation zones might depend on the antenna employed [4]

-When describing heat transfer, why did the authors decided to use Pennes bio-heat model among the ones proposed in literature through the years [5, 6]? Please report some discussion about this point here

-Did the authors here neglect perfusion dependence from the temperature [7]? If so, please report this as an assumption for the present computations

-In the reviewer's opinion, novelty from the present paper could be better described by improving the state of art in the introduction. Indeed, there are many contributions from the literature referred to the last years that haven't been considered in the literature review, with references to both modeling approach and novel solutions for MW ablation. Since in this contribution the authors focused on both aspects, some improvements in the manuscript are suggested. Microwave ablation have been extensively analyzed by including different aspects within the predictive models such as variable local properties [8], tissue dilation [9], as well as tissue shinkrage [10]; on the other hand, with references to the ablation procedures, pulsating heat sources [11], or number of slots employed [12], have been proposed through the years. Therefore, in order to underline what has been done up to now in the open literature, what should be done, and what is done here, the authors should improve their state of art based on these observations.

-A cell death model is presented here in a separate subsection. Why did the authors decided to use this approach instead of other ones, say CEM43, Arrenhius damage model, or temperature threshold criterion [5]? Please report here pro and cons of the present approach selected for cell death model

-It is not really clear how did the authors decided to locate the antenna within the liver. Which criterion did they use? As future plans, they could also report that some optimization of this procedure might be used by employing different location, timing, and positions, for the antenna. Optimizing ablation shape is still a challenging task to obtain the best ablated zone [13], and using this procedure might be really helpful

-Model validation is here shown with two different values of the r-coordinate. Are the experimental results [14] from ex-vivo or in-vivo experiments? Furthermore, even if the agreement looks fairly well, could the authors report some discussion about why for r = 4.5 mm the model underpredicts temperatures, while for r = 7 mm there is some overprediction?

-Within the paper, it is suggested to present some discussion about pro/cons between the present solution and a conventionally-used single-antenna solution

-Please improve figures. For instance, in Fig. 4, there are no axis and colorbar names here. Is this the heat generation in W/m3?

-In Fig. 8 the authors report temperature evolution through time. First of all please report the spatial location of the point investigated. Secondly, if the authors consider water phase change within their model, why the temperature after 100 °C still continues its growth? Some phase change might arise

-For Fig. 10, please report units (these should me millimeters)

[1] Ortega-Palacios, R., Trujillo-Romero, C. J., Cepeda Rubio, M. F. J., Vera, A., Leija, L., Reyes, J. L., ... & Vega-López, M. A. (2018). Feasibility of using a novel 2.45 GHz double short distance slot coaxial antenna for minimally invasive cancer breast microwave ablation therapy: Computational model, phantom, and in vivo swine experimentation. Journal of Healthcare Engineering, 2018.

[2] Guerrero Lopez, G. D., Cepeda Rubio, M. F. J., Hernandez Jacquez, J. I., Vera Hernandez, A., Leija Salas, L., Valdés Perezgasga, F., & Flores García, F. (2017). Computational FEM model, phantom and ex vivo swine breast validation of an optimized double-slot microcoaxial antenna designed for minimally invasive breast tumor ablation: theoretical and experimental comparison of temperature, size of lesion, and SWR, preliminary data. Computational and mathematical methods in medicine, 2017.

[3] Selmi, M., Bin Dukhyil, A. A., & Belmabrouk, H. (2019). Numerical analysis of human cancer therapy using microwave ablation. Applied Sciences, 10(1), 211.

[4] Hendriks, P., Berkhout, W. E. M., Kaanen, C. I., Sluijter, J. H., Visser, I. J., van den Dobbelsteen, J. J., ... & Burgmans, M. C. (2021). Performance of the Emprint and Amica Microwave Ablation Systems in ex vivo Porcine Livers: Sphericity and Reproducibility Versus Size. CardioVascular and Interventional Radiology, 44, 952-958.

[5] Andreozzi, A., Iasiello, M., & Tucci, C. (2020). An overview of mathematical models and modulated-heating protocols for thermal ablation. Advances in Heat Transfer, 52, 489-541.

[6] Nakayama, A., & Kuwahara, F. (2008). A general bioheat transfer model based on the theory of porous media. International Journal of Heat and Mass Transfer, 51(11-12), 3190-3199.

[7] Shi, J., Chen, Z., & Shi, M. (2009). Simulation of heat transfer of biological tissue during cryosurgery based on vascular trees. Applied Thermal Engineering, 29(8-9), 1792-1798.

[8] Tucci, C., Trujillo, M., Berjano, E., Iasiello, M., Andreozzi, A., & Vanoli, G. P. (2022). Mathematical modeling of microwave liver ablation with a variable-porosity medium approach. Computer Methods and Programs in Biomedicine, 214, 106569.

[9] Keangin, P., Wessapan, T., & Rattanadecho, P. (2011). Analysis of heat transfer in deformed liver cancer modeling treated using a microwave coaxial antenna. Applied Thermal Engineering, 31(16), 3243-3254.

[10] Farina, L., Weiss, N., Nissenbaum, Y., Cavagnaro, M., Lopresto, V., Pinto, R., ... & Goldberg, S. N. (2014). Characterisation of tissue shrinkage during microwave thermal ablation. International Journal of Hyperthermia, 30(7), 419-428.

[11] Andreozzi, A., Brunese, L., Iasiello, M., Tucci, C., & Vanoli, G. P. (2021). Numerical analysis of the pulsating heat source effects in a tumor tissue. Computer Methods and Programs in Biomedicine, 200, 105887.

[12] Keangin, P., Rattanadecho, P., & Wessapan, T. (2011). An analysis of heat transfer in liver tissue during microwave ablation using single and double slot antenna. International Communications in Heat and Mass Transfer, 38(6), 757-766.

[13] Trujillo, M., Prakash, P., Faridi, P., Radosevic, A., Curto, S., Burdio, F., & Berjano, E. (2020). How large is the periablational zone after radiofrequency and microwave ablation? Computer-based comparative study of two currently used clinical devices. International Journal of Hyperthermia, 37(1), 1131-1138.

[14] Yang, D., Converse, M. C., Mahvi, D. M., & Webster, J. G. (2007). Expanding the bioheat equation to include tissue internal water evaporation during heating. IEEE Transactions on Biomedical Engineering, 54(8), 1382-1388.

Reviewer #3: Authors report VFEM assisted computation of temperatures and FDM assisted computation of thermal damage zones for microwave assisted thermal ablation of liver tumor.

Authors mentioned that MWA of tumors larger than 3 cm to achieve spherical ablation region needs to be addressed. Though it is reported that MWA of tumor sizes 3-5 cm could be attained through repeated overlapping ablation with single antenna leading to elliptical ablation zone.

In this paper a spherical tumor of 4 cm diameter is considered for computations.

In my opinion, following comments need to be addressed before publication of this article.

1. Liver tumors are highly perfused, so dynamics of blood perfusion (temperture dependent) as well as heterogeneity of blood flow within the tumor should be considered to get realistic ablation zones. This becomes more important considering large size of the domain considered by the authors (8cm x 8cm).

2. Currently used Eq 11 accounts for just ON/OFF of blood perfusion instead of the complete spatiotemporal perfusion dynamics and this will not provide correct ablation zones. Authors may refer to literature Soni et al 2015 Int J Hyperthermia etc.

3. Numerical value of blood perfusion, considered for the simulation, could not be located in manuscript. Kindly mention this in Table 3 alongwith other parameters.

4. It is not clear, whether the electro-thermal properties corresponding to the ablated region (in total computational domain) are considered for computing the next ablated region, for sequential insertion of the electrode.

5. Please discuss whether the perfusion is cutoff for the first ablated region for computation of the second ablation zone (overlapping of ablation zones) for the next lectrode insertion point/time step and so on.

6. Instead of mentioning sequential insertion points/locations in the tumor, a normalized parameter (probe sepearation, time points) w.r.t. tumor volume may be provided to generalize the multiple elctrode insertion locations & exposure time steps based on a tumor volume.

7. Page 10, line 242 - Please justify the chosen locations for temperature measurements.

8. Page 10, line 253 - Authors mention good agreement with the published literature. But earlier it is mentioned that this literature reports for oval shaped ablation zones. So, how the claim of spherical zones is arrived. Fig. 6b also supports oval shaped ablation zone.

9. Table 4 - There are still remanant tumor cells for 4 minutes exposure as mentioned. Please discuss the strategy to avoid these.

10. Please label the axes in figures 4 and 6 as well as other such figures.

11. Fig. 5 - Please mention the dielectric and thermal parameters, in tablular form, for both the studies (experimental, numerical).

6. PLOS authors have the option to publish the peer review history of their article (what does this mean?). If published, this will include your full peer review and any attached files.

Reviewer #1: No

Reviewer #2: No

Reviewer #3: **Yes: **Sanjeev Soni

---

## [Author Response · Author response to Decision Letter 0]

5 Mar 2023

Thank you for asking us to submit revised paper. 

We answered the reviewer's questions. We modified the paper according to the reviewer's comments.

---

## [Decision Letter · Decision Letter 1]

2 May 2023

PONE-D-22-35134R13D Modeling of  Vector/Edge Finite Element Method for Multi-ablation  Technique for Large Tumor- Computational ApproachPLOS ONE

Dear Dr. boregwoda,

Thank you for submitting your manuscript to PLOS ONE. After careful consideration, we feel that it has merit but does not fully meet PLOS ONE’s publication criteria as it currently stands. Therefore, we invite you to submit a revised version of the manuscript that addresses the points raised during the review process.

We look forward to receiving your revised manuscript.

Kind regards,

Cihun-Siyong Alex Gong, Ph.D.

Academic Editor

PLOS ONE

Reviewers' comments:

Reviewer's Responses to Questions

**Comments to the Author**

1. If the authors have adequately addressed your comments raised in a previous round of review and you feel that this manuscript is now acceptable for publication, you may indicate that here to bypass the “Comments to the Author” section, enter your conflict of interest statement in the “Confidential to Editor” section, and submit your "Accept" recommendation.

Reviewer #2: (No Response)

Reviewer #4: (No Response)

Reviewer #5: (No Response)

2. Is the manuscript technically sound, and do the data support the conclusions?

Reviewer #2: Yes

Reviewer #4: Partly

Reviewer #5: Yes

3. Has the statistical analysis been performed appropriately and rigorously? 

Reviewer #2: N/A

Reviewer #4: N/A

Reviewer #5: Yes

4. Have the authors made all data underlying the findings in their manuscript fully available?

Reviewer #2: Yes

Reviewer #4: Yes

Reviewer #5: Yes

5. Is the manuscript presented in an intelligible fashion and written in standard English?

Reviewer #2: Yes

Reviewer #4: Yes

Reviewer #5: Yes

6. Review Comments to the Author

Reviewer #2: The revised version of the present manuscript looks far better than the first version; most of the questions from the reviewer have been properly answered. The manuscript in the present form presents interesting results for whom investigates thermal ablation, described via an innovative procedure based on the 3D vector finite element method. However, some minor improvements should be done by the authors, especially to underline why investigating this subject is still important and which is the novelty here, and which are the assumptions behind the present model. Please find in the following more details about these points.

-The authors report that they use a single-slot commercial coaxial antenna; in the rebuttal, the authors make references to studies from (doi.org/10.48550/arXiv.2008.02032, doi.org/10.1016/j.ijheatmasstransfer.2012.10.043) about this point. What did they mean when they wrote "this was used in one of the Author's previous projects for treating live patients [...]"? It is possible for the authors to report more details about it? For instance, are they talking about some clinical practice done via single-slot antenna? Furthermore - and far more important - please be sure that in Fig. 3 all the quotations have been included. For instance, which is the axial distance between the slot and the antenna's tip? Giving appropriate quotations for the geometry is fundamental for whom wants to reproduce the same model in the future

-In (doi.org/10.1016/j.ijheatmasstransfer.2012.10.043), it is true that the authors concluded that single-slot antennas might improve maximum SAR; on the other hand, it is also reported that the volumetric SAR is higher for the double slot MCA. Therefore, the authors should include all this when describing the geometrical model here, in order to underline that, even if some aspects might be better for double-slot antennas, the authors decided to use a single-slot antenna here

-The authors report in their rebuttal that they decide to use Pennes' bioheat model because liver parameters are homogeneous and isotropic, while blood vessels and flow might not be considered within the computational domain. That's seems unaccurate since, if one considers tissue plus blood as a heat exchanger, their heat capacities are not so different so one would expect an increase of arterial temperature within the perfusion term. The real reason of why Pennes bioheat equation might be still used nowadays, instead of other model like Klinger (doi.org/10.1007/BF02464617) or Chen and Holmes (doi.org/10.1111/j.1749-6632.1980.tb50742.x), is about its simplicity (doi.org/10.1016/bs.aiht.2020.07.003, doi.org/10.1016/j.ijheatmasstransfer.2007.05.030) - and this can be applied to the present study, but the author should report this as a limitation of the present model here.

-The authors done some improvements in the introduction to describe what has been done up to now, especially with references to cells death during thermal ablation (doi.org/10.1007/s10439-010-0177-1). However, it is still not clear for readers why researchers might be still interested in improving tumor thermal ablation modeling and standards; there is no mention in the introduction about modeling for local properties to be variable (doi.org/10.1016/j.cmpb.2021.106569), tissue dilation (doi.org/10.1016/j.applthermaleng.2011.06.005) or shinkrage (doi.org/10.3109/02656736.2014.957250), as well as standards improvements by using time-variable sources (doi.org/10.1016/j.cmpb.2020.105887) or different slots (doi.org/10.1016/j.ijheatmasstransfer.2012.10.043). Therefore, the paper introduction might be improved to underline that it is still important to investigate tumor thermal ablation from a research point of view

-The authors report that they neglect perfusion dependence vs temperature up to 60 °C. However, if one considers the perfusion rate vs temperature linear function reported in (doi.org/10.1016/j.applthermaleng.2008.08.014, doi.org/10.1016/j.applthermaleng.2011.06.005), by considering as an example 40 °C and 60 °C, perfusion rate might vary of about 25-30%. Therefore, the authors should include neglecting temperature dependence of perfusion as an assumption of their model here

Reviewer #4: 1.In Introduction, the frequency of 2.45 GHz is a common frequency in MWA but it is not the only one. A range of frequencies are addressed in previous studies.

2.Please describe the novelties and specific contribution of this work in the field in more detail at the last paragraph of Introduction.

3.Thermal properties of ablated zone change after MWA. Moreover, there is a time interval between changing the location of antenna in which the temperature of previously ablated reduce. Do the authors consider these two points? If not, how you can inform that the effect of these two parameters is negligible?

4.It suggested that the authors try the other treatment durations. For example, 5 min in two sections or 4 min in 3 parts. And find the optimum value for the relation between the MWA duration and the number of positions.

5.It suggested that the authors compare the treatment efficacy if their presented treatment strategy with the conventional MWA treatment which applied one time.

6.Can you find a more efficient treatment by changing and optimizing the positions of the antenna at each step?

7.It is suggested that the authors discuss tumor contraction in more detail, i.e., the effect, the importance, its impact on treatment outcomes, and its contribution to previous studies.

Reviewer #5: The manuscript contains many interesting results concerning microwave ablation. However, some issues should be addressed.

1)The following references should be cited

─Effects of target temperature on thermal damage during temperature-controlled MWA of liver tumor, 2022

─Modeling of heat transfer distribution in tumor breast cancer during microwave ablation therapy, 2022

─Numerical analysis of human cancer therapy using microwave ablation, 2021

2)How does the water contained in the body behave when the temperature exceeds 100°C?

3)Figure 8: Is the model still valid for all the temperatures?

7. PLOS authors have the option to publish the peer review history of their article (what does this mean?). If published, this will include your full peer review and any attached files.

Reviewer #2: No

Reviewer #4: No

Reviewer #5: No

---

## [Author Response · Author response to Decision Letter 1]

16 May 2023

Responses to the Reviewers’ Questions

We would like to thank all the reviewers for their valuable suggestions. We answered all the questions and modified the manuscript according to them. As one of the reviewers suggested, we simulated and presented the results for the following models: 1) Inserting antennas at three locations sequentially in the triangle pattern 2) producing heat at one location (single-ablation technique). 

Antenna separation distance plays an important role in creating an appropriate ablation zone. The relation between the antenna separation distance, heating time, and tumor volume for the multi-ablation technique is left for future investigation

Reviewer 2 

1] The authors report that they use a single-slot commercial coaxial antenna; in the rebuttal, the authors make references to studies from (doi.org/10.48550/arXiv.2008.02032, doi.org/10.1016/j.ijheatmasstransfer.2012.10.043) about this point. What did they mean when they wrote "this was used in one of the Author's previous projects for treating live patients "? It is possible for the authors to report more details about it? For instance, are they talking about some clinical practice done via single-slot antenna? Furthermore - and far more important - please be sure that in Fig. 3 all the quotations have been included. For instance, which is the axial distance between the slot and the antenna's tip? Giving appropriate quotations for the geometry is fundamental for whom wants to reproduce the same model in the future

ANS: One of the authors of this manuscript used a single-slot coaxial antenna in his previous projects, which accelerated treatment in real time using a GPU. The author has used the same antenna for experiments in the lab for that project. (https://www.gosmart-project.eu/index.html). Author has used same antenna in their previous work (axisymmetry 2d model) to analyse the impact of parametrs and pulstaing power on ablation zone( 2023 Jan;39(1):e3661. 

doi: 10.1002/cnm.3661. )

We included all the specifications and dimensions of the antenna in Figure 3, which will help to reproduce the results.

2] -In (doi.org/10.1016/j.ijheatmasstransfer.2012.10.043), it is true that the authors concluded that single-slot antennas might improve maximum SAR; on the other hand, it is also reported that the volumetric SAR is higher for the double slot MCA. Therefore, the authors should include all this when describing the geometrical model here, in order to underline that, even if some aspects might be better for double-slot antennas, the authors decided to use a single-slot antenna here

ANS: Thank you for your suggestions and included your points in the introduction. 

3] The authors report in their rebuttal that they decide to use Pennes' bioheat model because liver parameters are homogeneous and isotropic, while blood vessels and flow might not be considered within the computational domain. That seems unaccurate since, if one considers tissue plus blood as a heat exchanger, their heat capacities are not so different so one would expect an increase of arterial temperature within the perfusion term. The real reason of why Pennes bioheat equation might be still used nowadays, instead of other model like Klinger (doi.org/10.1007/BF02464617) or Chen and Holmes (doi.org/10.1111/j.1749-6632.1980.tb50742.x), is about its simplicity (doi.org/10.1016/bs.aiht.2020.07.003, doi.org/10.1016/j.ijheatmasstransfer.2007.05.030) - and this can be applied to the present study, but the author should report this as a limitation of the present model here.

ANS: We agree with your points, and mentioned in the manuscript. In our future work, we will use the topological geometry of the liver, which includes blood vessels, updated bio-heat equations, and fluid equations to analyse the velocity of the blood. 

4] The authors done some improvements in the introduction to describe what has been done up to now, especially with references to cells death during thermal ablation (doi.org/10.1007/s10439-010-0177-1). However, it is still not clear for readers why researchers might be still interested in improving tumor thermal ablation modeling and standards; there is no mention in the introduction about modeling for local properties to be variable (doi.org/10.1016/j.cmpb.2021.106569), tissue dilation (doi.org/10.1016/j.applthermaleng.2011.06.005) or shinkrage (doi.org/10.3109/02656736.2014.957250), as well as standards improvements by using time-variable sources (doi.org/10.1016/j.cmpb.2020.105887) or different slots (doi.org/10.1016/j.ijheatmasstransfer.2012.10.043). Therefore, the paper introduction might be improved to underline that it is still important to investigate tumor thermal ablation from a research point of view

ANS: The literature review in the introduction was improved as much as possible by the authors. The authors include details on the mathematical model, numerical approaches, significance of the vector finite element method, frequency range, constant and time-dependent input power, antenna design, tissue deformation, and a literature review on ablation zones using single ablation. And the author also explained the difficulties in treating large tumors using single ablation or multiple antennas. Then, finally, explain the novelty and multi-ablation technique and its importance.

5] The authors report that they neglect perfusion dependence vs temperature up to 60 °C. However, if one considers the perfusion rate vs temperature linear function reported in (doi.org/10.1016/j.applthermaleng.2008.08.014, doi.org/10.1016/j.applthermaleng.2011.06.005), by considering as an example 40 °C and 60 °C, perfusion rate might vary of about 25-30%. Therefore, the authors should include neglecting temperature dependence of perfusion as an assumption of their model here

ANS: Yes, the cut-off blood perfusion model is an assumption of this study, which we mentioned in the manuscript as well. 

Reviewer 4

1]. In the Introduction, the frequency of 2.45 GHz is a common frequency in MWA but it is not the only one. A range of frequencies is addressed in previous studies.

ANS: Yes, This study is not focused on the effect of frequency on the treatment since it has been well-established in the literature. And, also we mentioned the frequency range and optimum frequency used in microwave ablation in the introduction part. We used a fixed frequency of 2.45 GHz in our work since many researchers have been used and experimental results are available in the literature for this frequency [1].

[1] Yang, D., Converse, M.C., Mahvi, D.M. and Webster, J.G., 2007. Expanding the bioheat equation to include tissue internal water evaporation during heating. IEEE Transactions on Biomedical Engineering, 54(8), pp.1382-1388.

2]. Please describe the novelties and specific contribution of this work in the field in more detail in the last paragraph of the Introduction.

ANS: Thank you for your suggestion. We explained this in a detailed manner in the introduction.

3]. Thermal properties of ablated zone change after MWA. Moreover, there is a time interval between changing the location of the antenna in which the temperature of the previously ablated reduces. Do the authors consider these two points? If not, how you can inform that the effect of these two parameters is negligible?

ANS: The temperature in the ablated zone starts decreasing during the clinical time (the time interval required to change the antenna locations). Since clinical time is very short in reality, as a primary investigation, we assumed clinical time was zero in simulation. Moreover, the variation of temperature away from the antenna is very low; therefore, clinical time will not affect the boundary of the ablation zone.  

We simulated the ablation zone for a model for 3 minutes at each position and 1 minute as clinical time, and there is not much difference in the ablation zone (volume of the ablation zone with clinical time is 21.1 cm3, without clinical time is 20.9 cm3).  

4]. It suggested that the authors try the other treatment durations. For example, 5 min in two sections or 4 min in 3 parts. And find the optimum value for the relation between the MWA duration and the number of positions.

ANS: One of the difficulties in the multi-ablation technique is choosing the optimum value of antenna separation distance, number of antenna locations, and heating time, which depend on tumor size. One can optimize the efficiency of the mult-ablation technique by optimizing the above factors. The relationship between the separation distance, heating time, antenna locations, and tumor size for the multi-ablation technique is left for future investigation and we will communicate separately. As a primary investigation, we considered the tumor with a regular shape and tried to show the impact of inserting the antenna sequentially in the liver during the treatment on the ablation zone. 

As your suggestion, we simulated and presented the result for following settings. 

1) Antennas are inserted at four locations sequentially, with 3 and 4 minutes of heating time at each location.

2) Antennas are inserted at three locations sequentially in a triangle pattern, with 5 minutes of heating time at each location.

3) Antenna inserted at a single location, with heating time 20 min.

When the antenna was placed for 4 min at each position, 95.5% of the tumor cells and 1.8 cm3 of healthy cells were killed. 70% of cancer cells were killed when antennas were inserted at three locations sequentially with a heating time of 5 minutes at each location. The single ablation technique took 20 minutes to kill 95.97% of the tumor, and 7.76 cm3 of healthy cells were killed.

5]. It suggested that the authors compare the treatment efficacy of their presented treatment strategy with the conventional MWA treatment which is applied one time

ANS: We simulated results by an antenna at one position. The single-ablation technique took 20 minutes to kill 95.97% of tumors and it kills 7.76 cm3 healthy cells. Whereas the multi-ablation technique destroys 95.5% of tumors and kills 1.8 cm3 of healthy cells. 

6]. Can you find a more efficient treatment by changing and optimizing the positions of the antenna at each step?

ANS: The efficiency of the ablative treatment depends on the ablation zone created at the end of the treatment with minimum thermal damage to healthy cells. The shape and volume of the ablation zone of the multi-ablation technique depend on the antenna separation distance between the antennas and the heating time at each location. 

One can create a large ablation zone during the treatment by choosing the appropriate antenna separation distance and number of locations. Thermal damage can be optimised by optimising the heating time and locations of the antenna. And also, one can use the pulsating power and directed antennas to optimize the thermal damage.

As a primary investigation, we considered a regular-shaped tumor, and antennas were inserted in a square and equitriangle pattern. Optimising the number of locations, antenna separation distance, heating time, input power, and frequency is important in multi-ablation treatment to increase the efficiency of the treatment, which is kept for future investigation. 

7]. .It is suggested that the authors discuss tumor contraction in more detail, i.e., the effect, its importance, its impact on treatment outcomes, and its contribution to previous studies.

ANS: Thank you for your suggestion. We explained this point in the section’s result and tissue localized contraction model.

Reviewer 5. 

1] The following references should be cited

 ─Effects of target temperature on thermal damage during temperature-controlled MWA of liver tumor, 2022

─Modeling of heat transfer distribution in tumor breast cancer during microwave ablation therapy, 2022

─Numerical analysis of human cancer therapy using microwave ablation, 2021

ANS: Above mentioned papers have been cited in the proper places. 

2] How does the water contained in the body behave when the temperature exceeds 100°C?

ANS: Many researchers [1, 2, 3] have studied the remaining water content in the tissue during treatment; therefore, the author has not included it in the manuscript. We studied the water content in the tissue during the treatment and presented here. We measured the water content in the tissue at four positions: 

(-10, 7, 16.5), (10, 7, 16.5), (-10, -7, 16.5), and (10, -7, 16.5) which are 5 mm away from the antenna and parallel to the slot. Water vaporization in the tissue remained constant at temperatures below 80 °C. At temperatures over 80 °C, the water content begins to drop at an increasing rate to about one-half by 104 °C and exponentially decays as the temperature gets higher. 

[1] Selmi, Marwa, Abdul Aziz Bin Dukhyil, and Hafedh Belmabrouk. "Numerical analysis of human cancer therapy using microwave ablation." Applied Sciences 10, no. 1 (2019): 211.

[2] Yang, Deshan, Mark C. Converse, David M. Mahvi, and John G. Webster. "Measurement and analysis of tissue temperature during microwave liver ablation." IEEE transactions on biomedical engineering 54, no. 1 (2006): 150-155

[3] Yang, Deshan, Mark C. Converse, David M. Mahvi, and John G. Webster. "Expanding the bioheat equation to include tissue internal water evaporation during heating." IEEE Transactions on Biomedical Engineering 54, no. 8 (2007): 1382-1388.

3] Figure 8: Is the model still valid for all the temperatures?

ANS: Figure 8 represents the maximum temperature in the domains, which usually occurs near the slot or tip of the antenna. The variation in temperature near the antenna is greater, as can be seen from the experimental results [1]. As we move away from the antenna, the variation in temperature is less, and it may attain equilibrium. If we measure the temperature away from the antenna, it may not follow the Sudeen variation, rapid growth, and rapid decay in temperature.

---

## [Decision Letter · Decision Letter 2]

17 Jul 2023

3D Modeling of  Vector/Edge Finite Element Method for Multi-ablation  Technique for Large Tumor- Computational Approach

PONE-D-22-35134R2

Dear Dr. boregwoda,

We’re pleased to inform you that your manuscript has been judged scientifically suitable for publication and will be formally accepted for publication once it meets all outstanding technical requirements.

Kind regards,

Suhaib Ahmed, Ph.D.

Academic Editor

PLOS ONE

Additional Editor Comments (optional):

All the suggested changes have been incorporated in the revised manuscript and acknowledged by the reviewers.

Reviewers' comments:

Reviewer's Responses to Questions

**Comments to the Author**

1. If the authors have adequately addressed your comments raised in a previous round of review and you feel that this manuscript is now acceptable for publication, you may indicate that here to bypass the “Comments to the Author” section, enter your conflict of interest statement in the “Confidential to Editor” section, and submit your "Accept" recommendation.

Reviewer #1: All comments have been addressed

Reviewer #5: All comments have been addressed

2. Is the manuscript technically sound, and do the data support the conclusions?

Reviewer #1: Yes

Reviewer #5: Yes

3. Has the statistical analysis been performed appropriately and rigorously? 

Reviewer #1: I Don't Know

Reviewer #5: Yes

4. Have the authors made all data underlying the findings in their manuscript fully available?

Reviewer #1: Yes

Reviewer #5: Yes

5. Is the manuscript presented in an intelligible fashion and written in standard English?

Reviewer #1: Yes

Reviewer #5: Yes

6. Review Comments to the Author

Reviewer #1: 'Dear Editorial Board and Authors,

I am writing to express my appreciation for the efforts made in addressing my concerns regarding the manuscript submitted for publication. I am pleased to inform you that the authors have made significant improvements to their work.

Reviewer #5: All the comments have been accurately answered.

The English has been improved.

The references were completed.

Accept

7. PLOS authors have the option to publish the peer review history of their article (what does this mean?). If published, this will include your full peer review and any attached files.

Reviewer #1: No

Reviewer #5: No

---

## [Editor Report · Acceptance letter]

19 Jul 2023

PONE-D-22-35134R2 

3D Modeling of Vector/Edge Finite Element Method for Multi-ablation Technique for Large Tumor- Computational Approach 

Dear Dr. Boregowda:

I'm pleased to inform you that your manuscript has been deemed suitable for publication in PLOS ONE. Congratulations! Your manuscript is now with our production department. 

Kind regards, 

on behalf of

Dr. Suhaib Ahmed 

Academic Editor

PLOS ONE